# Joint Probabilistic-Nyquist Pulse Shaping for an LDPC-Coded 8-PAM Signal in DWDM Data Center Communications

Xiao Han [1,2,*], Mingwei Yang [1], Ivan B. Djordjevic [1], Yang Yue [2], Qiang Wang [2], Zhen Qu [2] and Jon Anderson [2]

1   ECE Department, University of Arizona, Tucson, AZ 85721, USA; mingweiyang@email.arizona.edu (M.Y.); ivan@email.arizona.edu (I.B.D.)
2   Juniper Networks, 1133 Innovation Way, Sunnyvale, CA 94089, USA; yyue@juniper.net (Y.Y.); qiwang.thresh@gmail.com (Q.W.); zqu@juniper.net (Z.Q.); jonanderson@juniper.net (J.A.)
*   Correspondence: xhan322@email.arizona.edu

**Abstract:** M-ary pulse-amplitude modulation (PAM) meets the requirements of data center communication because of its simplicity, but coarse entropy granularity cannot meet the dynamic bandwidth demands, and there is a large capacity gap between uniform formats and the Shannon limit. The dense wavelength division multiplexing (DWDM) system is widely used to increase the channel capacity, but low spectral efficiency of the intensity modulation/direct detection (IM/DD) solution restricts the throughput of the modern DWDM data center networks. Probabilistic shaping distribution is a good candidate to offer us a fine entropy granularity and efficiently reduce the gap to the Shannon limit, and Nyquist pulse shaping is widely used to increase the spectral efficiency. We aim toward the joint usage of probabilistic shaping and Nyquist pulse shaping with low-density parity-check (LDPC) coding to improve the bit error rate (BER) performance of 8-PAM signal transmission. We optimized the code rate of the LDPC code and compared different Nyquist pulse shaping parameters using simulations and experiments. We achieved a 0.43 dB gain using Nyquist pulse shaping, and a 1.1 dB gain using probabilistic shaping, while the joint use of probabilistic shaping and Nyquist pulse shaping achieved a 1.27 dB gain, which offers an excellent improvement without upgrading the transceivers.

**Keywords:** pulse amplitude modulation; nyquist pulse shaping; DWDM system; LDPC coding

## 1. Introduction

In view of the current development of the annual growth rate of data center transmission, the widely used coherent optical communication [1–5] is moving toward the data center networks market, but has not dominated because of its high cost, high power consumption, and implementation complexity. Self-coherent detection is a viable solution to reduce the cost, but its expensive coherent receiver still limits its use to a narrow range of applications [6–8]. To match the low-cost requirement, many researchers have paid increasing attention to pulse-amplitude modulation (PAM) [9–11]. However, for the currently widely used uniform distribution, coarse entropy granularity of M-ary PAM (M = 2, 4, 8, . . . ) cannot meet the dynamic bandwidth demands. More importantly, there exists a large capacity gap between the uniform modulation formats and the Shannon limit.

To compensate for the performance loss, in recent years, a constellation shaping scheme has attracted increasing research attention, which include geometric shaping (GS) [12–17], probabilistic shaping (PS) [18–23], and hybrid geometric-probabilistic shaping [24–27]. These different shaping

schemes can approach the Shannon limit. Since it may be easier to have a common standard agreement among network service providers if the PS scheme is used for constellation shaping, PS should be more suitable to be used in data center networks. PS can be realized using Huffman coding [28,29], many-to-one mapping [30], and a constant composition distribution matcher (CCDM) [31,32]. Given that a CCDM can flexibly generate fractional entropy without systematic error, the optical industry is more inclined to apply a CCDM-based PS scheme. PS can not only be used for quadrature amplitude modulation (QAM) formats, but also for a PAM scheme [33–35]. For a PAM scheme, it imposes an exponential-like distribution on a set of equidistant constellation points. It transmits symbols with smaller amplitudes more often than larger ones, which can offer us a fine entropy granularity and enable a transmission with a lower signal-to-noise ratio (SNR) at the same forward error correction (FEC) overhead.

The requirement of channel capacity is greatly increasing nowadays, which leads to the dense wavelength division multiplexing (DWDM) system being widely using in data center networks [36–38], but the low spectral efficiency of the intensity modulation/direct detection (IM/DD) solution restricts the throughput of the modern DWDM data center networks, which results in large investments to upgrade the transceivers in order to meet the increasing bandwidth requirement.

Nyquist pulse shaping (NPS) is a good solution to increase the spectral efficiency of a DWDM system [39,40]; it uses a raised-cosine filter to limit the effective bandwidth and can reduce the inter-symbol interference (ISI) by properly selecting the roll-off factor (ROF) for NPS.

The purpose of this paper was to demonstrate the joint usage of both the PS distribution and NPS in a DWDM system for short-reach applications, in particular data center networks. In addition to the joint shaping format, the employment of a suitable FEC code was also important for improving the overall performance. Low-density parity-check (LDPC) codes represent excellent FEC candidates to be applied together with the proposed shaping scheme. We transmitted LDPC-coded 8-PAM signals for both uniform and nonuniform signaling and compared the performance for different LDPC code rates. We further evaluated the performance improvements when NPS was used for different ROFs in both PS and uniform distribution-based systems. We experimentally evaluated the bit error rate (BER) performance improvement of the proposed joint shaping scheme, compared with the uniform signaling scheme.

The rest of the paper is organized as follows. In Section 2, we introduce the DWDM system employing the proposed joint probabilistic-Nyquist pulse shaping scheme. In Section 3, we demonstrate the improvements with respect to uniform signaling via simulation and experimental verifications. Relevant concluding remarks are provided in Section 4.

## 2. Proposed PS-NPS-8-PAM-Based DWDM System

Figure 1 shows the proposed LDPC-coded PS-NPS-8-PAM signal generator. The input was a pseudorandom binary sequence (PRBS), and after the distribution matcher (DM), there was an array of different amplitudes that satisfy a certain probability distribution. After binary labelling and LDPC encoding [36], we obtained a block of LDPC-coded binary bits. After mapping to the PAM constellation points, we performed the Nyquist pulse shaping, followed by the DWDM multiplexing.

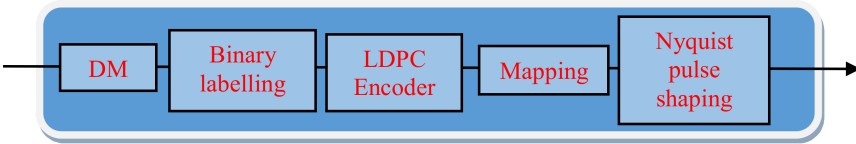

**Figure 1.** Low-density parity-check (LDPC)-coded, probabilistic shaping, Nyquist pulse shaping, 8-level pulse-amplitude modulation (PS-NPS-8-PAM) signal generator. DM: Distribution matcher.

The PS distribution performances are highly dependent on a selected distribution function. In this paper, we used an exponential distribution [33], in which different constellation points $a_i$ were transmitted with probabilities determined using:

$$P(a_i) = \exp(-\lambda\|a_i\|)/Z(\lambda), \ \lambda \geq 0, \tag{1}$$

where $Z(\lambda)$ is the normalization function used to ensure that the probabilities of occurrence of symbols sum up to one, which means $Z(\lambda)$ is defined as:

$$Z(\lambda) = \sum_i \exp(-\lambda\|a_i\|). \tag{2}$$

Nyquist pulse shaping is widely used in DWDM systems to improve the spectral efficiency [40]. It often uses a low-pass raised-cosine (RC) filter with the frequency response being:

$$H_{RC}(\omega) = \begin{cases} T_s & 0 \leq |\omega| < \pi(1-\text{ROF})/T_S \\ \frac{T_s}{2}\left(1 - \sin\left[\frac{T_S}{2\times\text{ROF}}\left(|\omega| - \frac{\pi}{T_s}\right)\right]\right), & \pi(1-\text{ROF})/T_S \leq |\omega| < \pi(1+\text{ROF})/T_S \\ 0 & |\omega| > \pi(1+\text{ROF})/T_S \end{cases} \tag{3}$$

where $\omega$ is the angular frequency, $T_s$ is the symbol duration, and ROF is the roll-off factor mentioned before. A small ROF value can make the frequency response an almost rectangular shape and be able to reduce the channel spacing but comes with a longer memory length and a higher generation complexity. In this study, we used the square-root raised cosine (SRRC) filter, whose transfer function is given as:

$$H_{SRRC}(\omega) = \sqrt{H_{RC}(\omega)}. \tag{4}$$

Figure 2 shows the frequency response for two different ROF values of neighboring DWDM channels, with the channel spacing set to 50 GHz. It is clear that a smaller ROF had a better spectral efficiency with a lower inter-channel crosstalk.

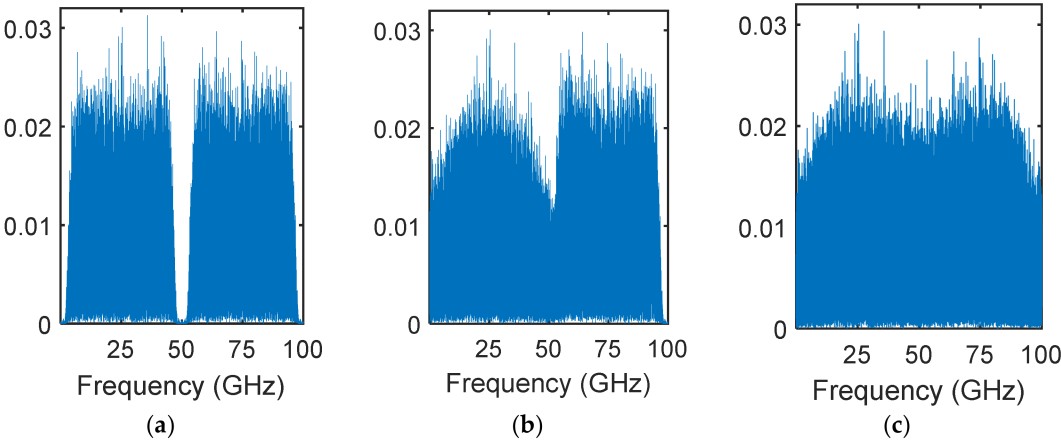

**Figure 2.** Frequency response with different roll-off factors (ROFs): (**a**) ROF$_1$ = ROF$_2$ = 0.1; (**b**) ROF$_1$ = 1, ROF$_2$ = 0.1; and (**c**) ROF$_1$ = ROF$_2$ = 1.

Figure 3 shows the whole DWDM system. On the transmitter side, we generated the PS-NPS-8-PAM signals using the generator in Figure 1 and sent them to the modulator for each channel with a different frequency. Then, after transmission, at the receiver, we used a super Gaussian filter to select every target frequency (wavelength).

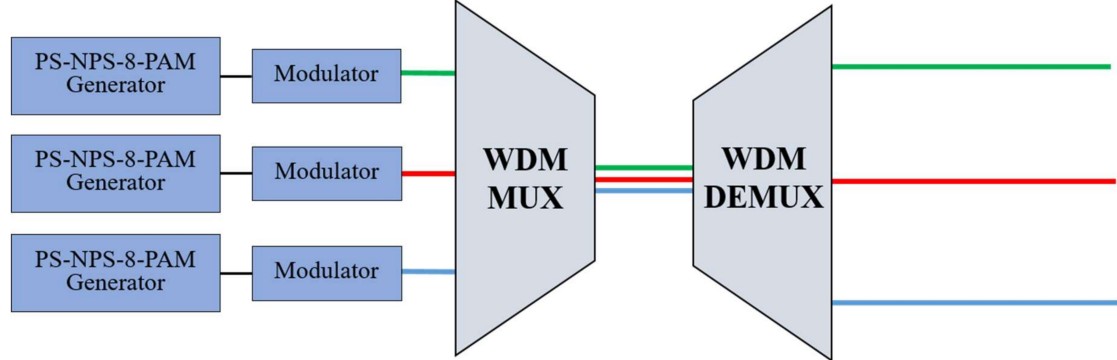

**Figure 3.** Structure of the PS-NPS-8-PAM-based DWDM system. DEMUX: demultiplexer.

## 3. Simulation and Data Center Experimental Results

In this section, we first describe the optimization of the code rate of the LDPC encoder for the PS distribution and show a different ROF factor performance using simulations. Then, we introduce the experimental setup and the improvement we achieved from our PS-NPS-8-PAM scheme.

### 3.1. Simulation Results

Figure 4 shows the BER performance comparison for LDPC-coded 8-PAM signals with PS and uniform distributions for different LDPC code rates. As the PS distribution has a smaller entropy, we needed to make sure that each modulation scheme had the same information rate to guarantee that the comparison was fair; in other words, they had the same FEC overhead. The code rate of the uniform distribution was 0.6, so the information rate was 1.8 bits/symbol, with a 66.7% FEC overhead. From Figure 4, we can see that all PS distributions outperformed the uniform distribution, and the best improvement offered a 0.8 dB signal-to-noise ratio (SNR) gain over the uniform distribution at a BER of $10^{-5}$, which appeared when code rate ($r$) equal to 0.8.

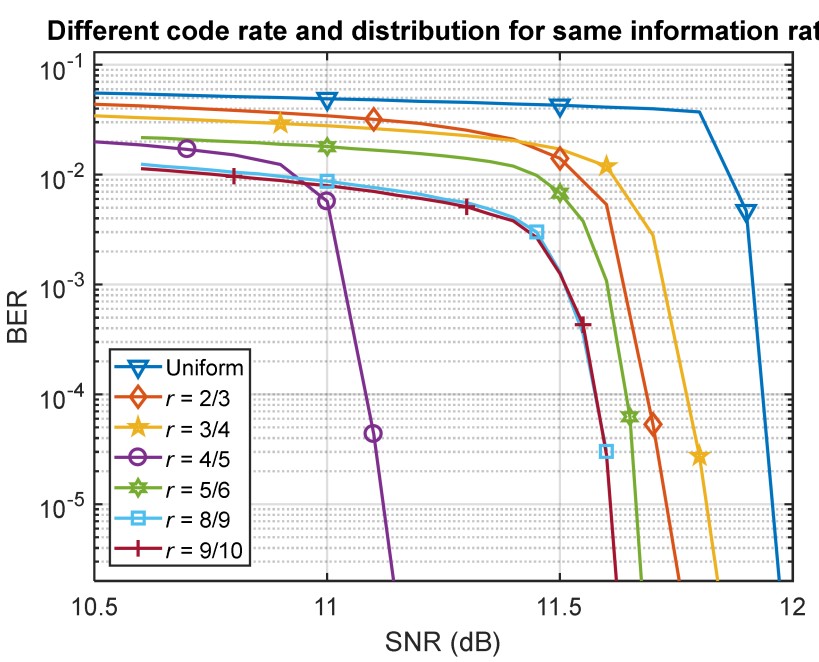

**Figure 4.** Bit error ratio (BER) performance for different LDPC code rates ($r$). SNR: Signal-to-noise ratio.

Figure 5 shows the BER performance of the LDPC-coded PS and uniform distribution for different values of the ROF. We can see for both distribution schemes that a smaller ROF achieved a better

BER performance, which was caused by the reason shown in Figure 2: a smaller ROF value can lead to the frequency response of a rectangular shape, improving the spectral efficiency. For every ROF value considered, we found that the PS distribution-based scheme always outperformed uniformly distribution-based one.

Figure 5a shows a very clear error flow when the ROF was set to 0.5. Namely, every LDPC code had an effective SNR region, in which it sufficiently improved the BER performance after a corresponding SNR threshold. According to Figure 2, a larger ROF brought a wider bandwidth and more inter-channel crosstalk, which prevented us reaching the SNR threshold of the employed LDPC code.

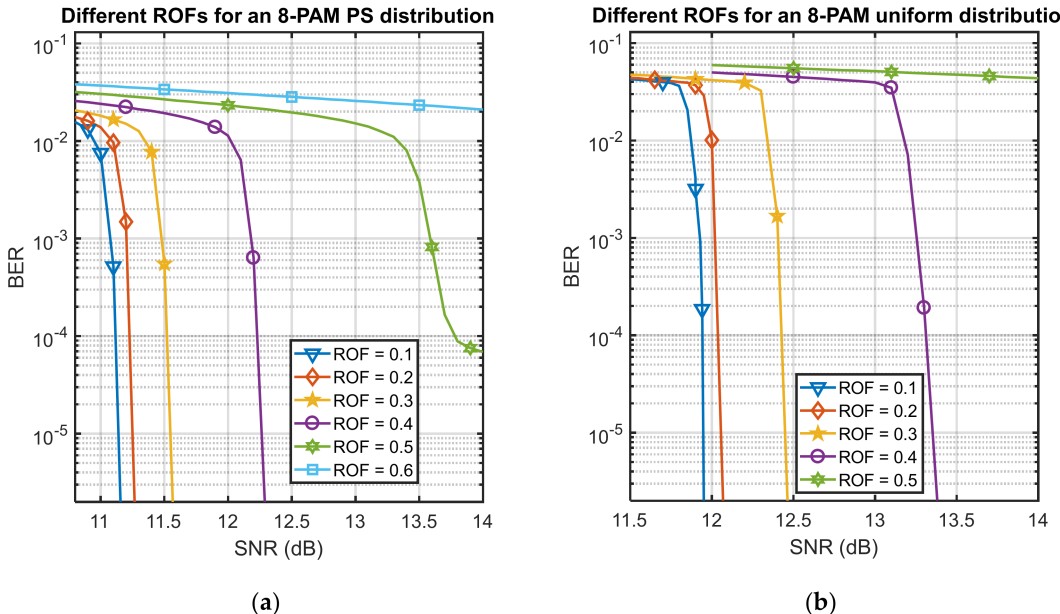

(**a**)　　　　　　　　　　　　　　　　　　　　　(**b**)

**Figure 5.** BER performance for different ROF values for: (**a**) a PS distribution with an LDPC code of rate $r = 0.8$ and (**b**) a uniform distribution with an LDPC code of rate $r = 0.6$.

*3.2. Experimental Setup and Results*

The performance of the LDPC-coded PS-NPS-8-PAM transmission in a fiber channel-based DWDM system was experimentally verified with the testbed depicted in Figure 6. Three 10 kHz-linewidth, continuous-wave, tunable sources (with the following center frequencies: $f_1 = 193.30$ THz, $f_2 = 193.35$ THz, and $f_3 = 193.40$ THz) were coupled by an optical coupler and launched to a Mach-Zehnder modulator. The PRBS was sent to our LDPC-coded PS-NPS-8-PAM generator. Then, the LDPC-coded PS-NPS signals were sent to an arbitrary waveform generator (AWGen) to create 11.5 GBaud 8-PAM signals with a 66.7% FEC overhead, which meant the bit rate was 20.7 Gbps. After being converted to the optical domain by the modulator, the resulting signals were boosted by an erbium-doped fiber amplifier (EDFA) with a 6 dB noise figure. The enhanced signal was then sent to a $1 \times 3$ coupler to be split and interleaved into three fibers with different lengths (delay lines). The corresponding outputs were then applied to a $32 \times 32$ arrayed waveguide grating (AWG)-based datacenter network at three different input ports. Every input port worked as a selective bandpass filter. At the output port, we obtained three different center frequencies with different delays, which worked as our DWDM system. The output signal was then mixed with an amplified spontaneous emission (ASE) noise signal using a $2 \times 2$ coupler. We also employed a variable optical attenuator (VOA) after the ASE noise source to emulate the different optical SNR (OSNR) channel conditions. At the receiver side, the targeted frequency $f_2$ was selected by a tunable filter (TF) and detected using a photodetector (PD). To collect the received baseband signal, we employed a 100 GS/s digital phosphor oscilloscope from Tektronix. Then, we performed offline digital signal processing (DSP) with the collected signals.

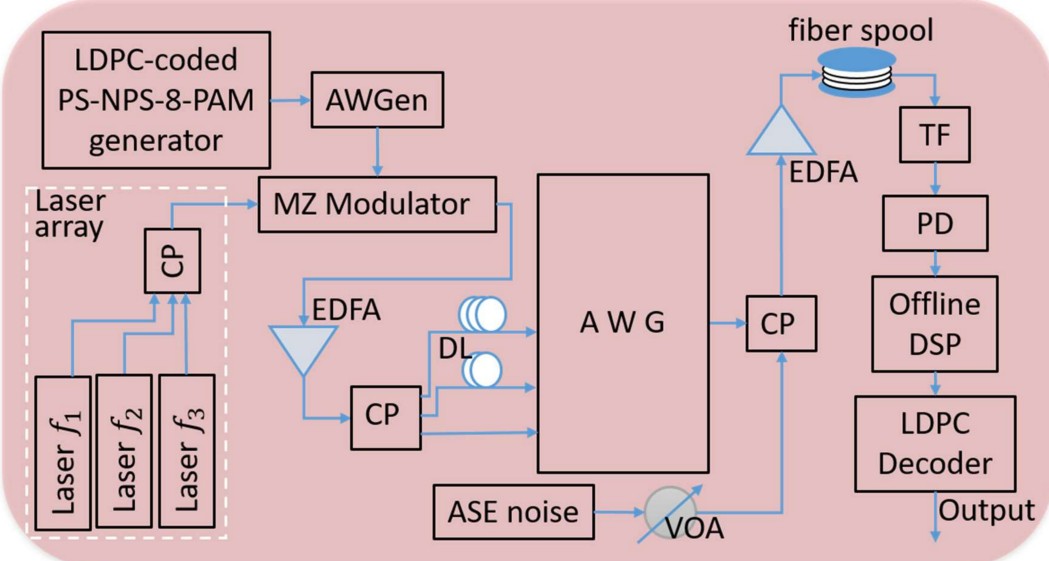

**Figure 6.** Experimental data center setup. CP: coupler, DL: delay line.

The BER performance of the LDPC-coded PS-NPS-8-PAM is shown in Figure 7a. We transmitted the PS-NPS-8-PAM signals with ROF = 0.8 and 1, and compared them against a PS-8-PAM. From this figure, we can see that when ROF = 1, it performed the same as the PS-8-PAM without NPS, which is because ROF = 1 will not change the pulse shape. On the other hand, when we set the ROF to 0.8, we obtained a 0.43 dB shaping gain improvement in OSNR at BER = $10^{-5}$. In Figure 7b, we compare the LDPC-coded PS- and uniform distribution-based schemes with both ROFs set to 0.8. We can see that uniform distribution required higher OSNRs, and we obtained a 1.1 dB shaping gain improvement in OSNR at BER = $10^{-5}$.

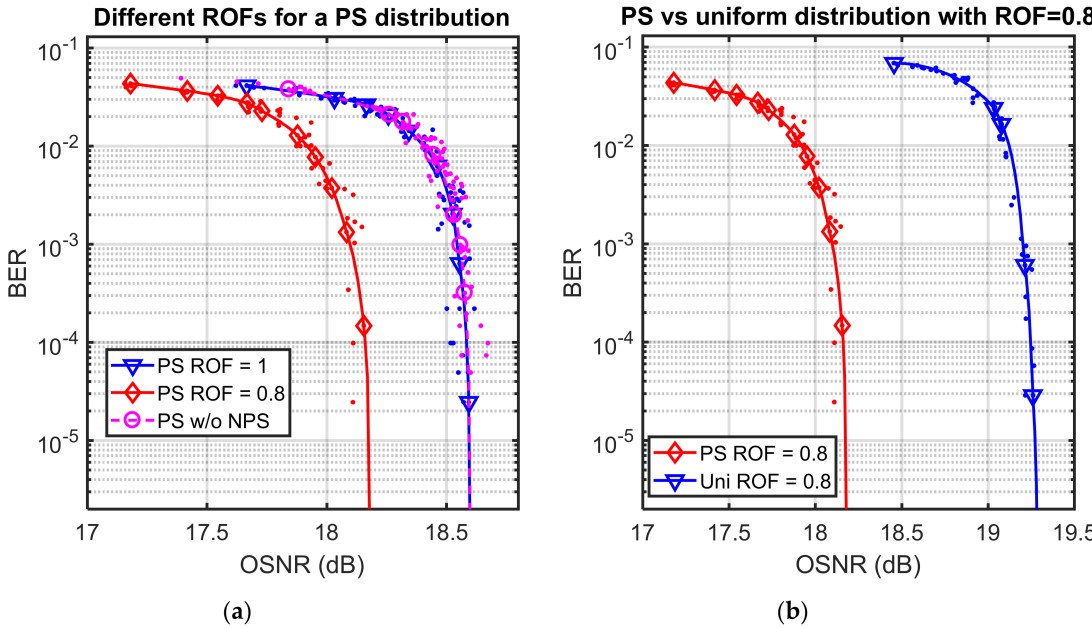

**Figure 7.** BER performance for: (**a**) a PS distribution with different ROFs and without NPS, and (**b**) a PS and uniform distribution comparison with the same ROF. OSNR: optical signal-to-noise ratio.

Figure 8 shows the comparison of BER performance between the LDPC-coded PS-NPS-8-PAM and LDPC-coded uniform distributed 8-PAM schemes. Evidently, the joint usage of PS and NPS could obtain a 1.27 dB OSNR improvement at BER = $10^{-5}$ compared to the uniform signaling.

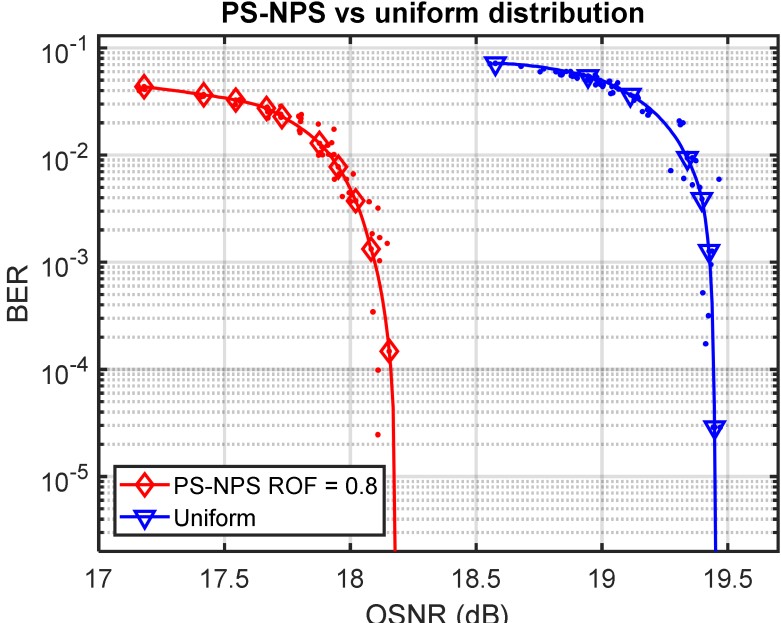

**Figure 8.** BER performance of an LDPC-coded PS-NPS-8-PAM against an LDPC-coded uniform 8-PAM.

## 4. Conclusions

In our simulation results, we found that a smaller ROF could provide a better BER performance, and to optimize the PS performance, we needed to consider both the signal entropy and the error correction capability of an LDPC code. In practical usage, there will be many potential limits on the ROF and FEC overhead, since in most cases, we can only use an appropriate ROF for a certain FEC overhead. We have shown that the optimization was not highly sensitive to either the ROF or FEC overhead, and we could do the optimization on any limit of the ROF and FEC overhead. The improvement can be achieved without any equipment upgrade, which means that the proposed LDPC-coded PS-NPS-8-PAM scheme can be widely applied in data center communications and other short-reach applications. We have shown that the Nyquist pulse shaping could provide a 0.43 dB improvement, and the probabilistic shaping provided a 1.1 dB gain. By the joint use of probabilistic shaping and Nyquist pulse shaping, we obtained a 1.27 dB performance gain.

**Author Contributions:** This research was conducted by X.H. M.Y. offered experimental support. Y.Y., Q.W., Z.Q. and J.A. supervised the NPS and PS idea and simulation during Han's internship at Juniper Networks. Both the simulation and the experimental demonstration were supervised by I.B.D.

**Funding:** This research was funded by Juniper Networks and NSF.

**Conflicts of Interest:** The authors declare no conflict of interest.

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
