# Peer review of "Joint Probabilistic-Nyquist Pulse Shaping for an LDPC-Coded 8-PAM Signal in DWDM Data Center Communications"

_applsci, doi:10.3390/app9234996_

Round 1

Reviewer 1 Report

A joint Probabilistic Nyquist Pulse Shaping for LDPC-coded 8-PAM Signal in DWDM Data Center Communication is proposed in the research. The authors provide a methodology to joint usage of probabilistic shaping and Nyquist pulse shaping with low density parity check LDPC coding to improve the BER performance of 8-PAM signal transmission.

The article is generally well written and structured. However, the clarity can be further improved by fixing some grammatical mistakes. e.g. can't, hasn't, What's etc. In references, some years are bold while others are normal.

Author Response

"The article is generally well written and structured. However, the clarity can be further improved by fixing some grammatical mistakes. e.g. can't, hasn't, What's etc. In references, some years are bold while others are normal."

Thank you for the review and the comments, we have corrected such grammatical mistakes, and in reference we bold all year. Anything need to be updated feel free to tell us, thank you.

Reviewer 2 Report

The article is well written. The results and conclusions are well presented and described. Here are some minor comments:

line 34: What's -> What is
lines 107 and 108: between section and subsection there should be some text.

line 138: The manuscript does not clearly state what kind of signals in DC technology (FC or GbE) and what bit rate (e.g., 10 or 100G) of DWDM signals were tested.

And a remark about the references:

I acknowledge that Authors have the significant contribution to the field, however, I find it difficult to accept that 20 out of 40 references are self-citations. I therefore strongly recommend that the number of self-citations is significantly reduced. My recommendation would be that not more than 20% of all references are self-citations.

Author Response

"line 34: What's -> What is
lines 107 and 108: between section and subsection there should be some text.

line 138: The manuscript does not clearly state what kind of signals in DC technology (FC or GbE) and what bit rate (e.g., 10 or 100G) of DWDM signals were tested.

And a remark about the references:

I acknowledge that Authors have the significant contribution to the field, however, I find it difficult to accept that 20 out of 40 references are self-citations. I therefore strongly recommend that the number of self-citations is significantly reduced. My recommendation would be that not more than 20% of all references are self-citations."

Now we have corrected such mistake in line 34 and similar ones in other place;

We have add a brief description between section and subsection lines 107 and 108;

In line 138, the experiment is fiber channel-based, and the bit rate is  20.7Gbps;

Now we have reduced the self-citations totally not more than 10, several of them are our previous work 

Anything need to be updated feel free to tell us, thank you for the review and comments!